# Micro-Doppler Signature Detection and Recognition of UAVs Based on OMP Algorithm

**DOI:** 10.3390/s23187922

**Published:** 2023-09-15

**Authors:** Shiqi Fan, Ziyan Wu, Wenqiang Xu, Jiabao Zhu, Gangyi Tu

**Affiliations:** Department of Electronics and Information Engineering, Nanjing University of Information Science and Technology, Nanjing 210044, China; 18726625717@163.com (S.F.); wzyann2022@163.com (Z.W.); wenqiangnuist@163.com (W.X.); zhujiabao0106@163.com (J.Z.)

**Keywords:** radar signal processing, identification of UAV, OMP algorithm, clutter suppression, micro-Doppler

## Abstract

With the proliferation of unmanned aerial vehicles (UAVs) in both commercial and military use, the public is paying increasing attention to UAV identification and regulation. The micro-Doppler characteristics of a UAV can reflect its structure and motion information, which provides an important reference for UAV recognition. The low flight altitude and small radar cross-section (RCS) of UAVs make the cancellation of strong ground clutter become a key problem in extracting the weak micro-Doppler signals. In this paper, a clutter suppression method based on an orthogonal matching pursuit (OMP) algorithm is proposed, which is used to process echo signals obtained by a linear frequency modulated continuous wave (LFMCW) radar. The focus of this method is on the idea of sparse representation, which establishes a complete set of environmental clutter dictionaries to effectively suppress clutter in the received echo signals of a hovering UAV. The processed signals are analyzed in the time–frequency domain. According to the flicker phenomenon of UAV rotor blades and related micro-Doppler characteristics, the feature parameters of unknown UAVs can be estimated. Compared with traditional signal processing methods, the method based on OMP algorithm shows advantages in having a low signal-to-noise ratio (−10 dB). Field experiments indicate that this approach can effectively reduce clutter power (−15 dB) and successfully extract micro-Doppler signals for identifying different UAVs.

## 1. Introduction

Recently, with the ubiquitous use of commercial and military unmanned aerial vehicles (UAVs), the country and the public are increasingly concerned about the privacy and security issues that come with them, which can be reflected in the demands for the revision and improvement of the relevant regulations on the control of UAVs at low altitudes [1,2]. The threat posed by unsupervised UAVs to urban security may be enormous, such as causing large airport operations to be paralyzed. The identification of UAVs is a significant part of their regulation, and the accurate acquisition of their characteristics is crucial. As a micro dynamic effect of small targets, the micro-Doppler frequency shift caused by rotor rotation can be used as an essential reference for calculating the rotational velocity of UAV rotor blades [3]. After obtaining a certain number of estimated parameters from the micro-Doppler features, unknown UAVs can be identified and classified according to these parameters [4,5,6].

In order to observe the micro-Doppler effect of UAV rotor blades and extract characteristic parameters, a theoretical micro-Doppler echo model of UAV rotor blades in radar detection systems was firstly established [7]. Time–frequency analysis is a common method to extract features from echoes, including short time Fourier transform (STFT) [8], Gabor transform [9], Wigner-Ville distribution (WVD) [10] and wavelet transform [11]. STFT adopts the method of intercepting the time window. However, designing the window function span correctly is not easy because it has poor resolution in the time domain on long time windows and poor resolution in the frequency domain on short time windows. The Gabor transform is optimized from STFT, and its window function is fixed as a Gaussian window, which makes its comprehensive resolution in time domain and frequency domain reach the optimum. Nonetheless, when it comes to abrupt and non-stationary signals, the Gabor transform will struggle to produce good results. For WVD, when faced with multi-component linear frequency modulation signals, their time–frequency resolution will decrease, and the presence of cross terms may lead to a decrease in accuracy. And the drawback of wavelet transform is its lack of self-adaptability. From this, it can be concluded that extracting micro-Doppler parameters directly from weak UAV echo signals with strong clutter interference through time–frequency analysis is challenging. Therefore, clutter suppression of the echo signal received by the radar before parameter estimation is a more suitable processing method.

Using cleverly designed filters is a common way to suppress clutter because they are simple and effective [12,13]. The Wiener filter is a popular noise reduction technique in modern signal processing [14,15]. However, additional knowledge about the signal or noise is needed to achieve optimal noise reduction because the use of this technology may delete components carrying crucial information. Furthermore, signal decomposition and reconstruction is a typical method for signal feature analysis [16]. The empirical mode decomposition (EMD) algorithm decomposes a signal into a series of intrinsic mode functions (IMF) [17,18], which can describe the local features of the signal. This algorithm has adaptability, and it decomposes the signal based on the time scale characteristic of the data itself without setting any basis function in advance. Nonetheless, the EMD algorithm also has some issues that need to be addressed, and spectrum aliasing is one of them [19,20,21]. When the signal energy from clutter and noise is much greater than that from the target echo, the EMD algorithm often finds it hard to extract useful target signals.

In addition, matching pursuit (MP) has been proven to be another effective signal decomposition method. It can decompose signals into the combination of basic wave forms in the dictionary, which is suitable for the reconstruction of different types of micro-Doppler signals [22,23]. The orthogonal matching pursuit (OMP) algorithm [24,25,26] is a greedy algorithm developed on the basis of MP, which has received widespread attention from researchers due to its low mathematical complexity and fast convergence speed [27]. However, when it comes to engineering implementation, a sufficient number of prior conditions are needed to form a complete dictionary for decomposing signals [28,29,30], without which it is nearly impossible to know the dispersion of signals.

In this paper, a new clutter reduction method is proposed to better apply the OMP algorithm to radar detection systems. This method is based on the decomposition and reconstruction of the calculated signal in the dictionary, which is constructed by the prior theoretical non-detection point echoes and the ground clutter received by the radar. The OMP algorithm assists in finding a clearer basis containing the target signal for subsequent time–frequency analysis. Through the flicker phenomenon and frequency shift of UAV rotor blades in the time–frequency domain, the length and speed of the rotor blades can be evaluated, and the unknown UAV can be further recognized and, finally, classified.

The remainder of this paper is organized as follows. In Section 2, an echo signal model of the UAV rotor blades is established based on the LFMCW radar, which lays a foundation for the construction of a relatively complete dictionary. Section 3 proposes a new method for constructing a clutter suppression dictionary based on the OMP algorithm and analyzes its feasibility, and the specific description of this method is also provided. In Section 4, the results of simulations and field tests are given and discussed. Finally, Section 5 summarizes the conclusions.

## 2. Echo Signal Model

### 2.1. Micro-Doppler Echo Model

The estimation of UAV rotor blade speed in the micro-Doppler signature requires the establishment and analysis of a theoretical echo model. When a LFMCW radar is used to detect the target, the distance between it and the target can be calculated by the frequency difference between the transmitted signal and the reflected signal, and its transmitted signal within a transmission cycle can be represented in the time domain as [7]
(1)$$S_{T}\left(t\right)=A_{0}\cos\left(2\pi \left(f_{0}t+\frac{1}{2}Kt^{2}\right)+\varphi_{0}\right)$$
where $A_{0}$ is the initial amplitude of $S_{T}\left(t\right)$; $f_{0}$ is the carrier frequency; K=B/T is the frequency modulation index, *B* is the effective bandwidth of $S_{T}\left(t\right)$, *T* is the emission period and $\varphi_{0}$ is the initial phase. So, the phase information expression of the transmitted signal can be written as
(2)$$P_{T}\left(t\right)=2\pi \left(f_{0}t+\frac{1}{2}Kt^{2}\right)+\varphi_{0}$$

When the detection target is in motion, due to the component of its velocity in the direction of the radar line of sight (LOS), the difference frequency obtained will no longer be a fixed constant. If the target is moving towards the radar, the delay between the transmitted signal and the echo signal can be approximated as
(3)$$\tau_{1}\left(t\right)\approx \frac{2(R_{0}-vt)}{c}$$
where *v* is the radial velocity of the target relative to the radar; $R_{0}$ is the initial distance between the target and radar; *c* is the speed of light. So, the echo signal received by the radar can be represented in the time domain as
(4)$$S_{R}\left(t\right)=K_{r}A_{0}\left[\cos2\pi \left(f_{0}\left(t\;-\;\tau_{1}\left(t\right)\right)\;+\;\frac{1}{2}K{\left(t\;-\;\tau_{1}\left(t\right)\right)}^{2}\right)\;+\;\varphi_{0}\right]$$
where $K_{r}$ is the loss coefficient and $\tau_{1}\left(t\right)$ is the time delay between transmitted and received signals at time *t*. And the phase information of the echo signal can be represented as
(5)$$P_{R}\left(t\right)=2\pi \left(f_{0}\left(t-\tau_{1}\left(t\right)\right)+\frac{1}{2}K{\left(t-\tau_{1}\left(t\right)\right)}^{2}\right)+\varphi_{0}$$

By combining Equations (Equation 2) and (Equation 5), the phase of baseband signal after mixing can be written as
(6)$$\begin{array}{cc} P_{b}\left(t\right) & =P_{R}\left(t\right)-P_{T}\left(t\right)\\ \; & =\left[2\pi \left(f_{0}\left(t-\tau_{1}\left(t\right)\right)+\frac{1}{2}K{\left(t-\tau_{1}\left(t\right)\right)}^{2}\right)+\varphi_{0}\right]-\left[2\pi \left(f_{0}t+\frac{1}{2}Kt^{2}\right)+\varphi_{0}\right]\\ \; & =2\pi [\frac{1}{2}K\tau_{1}{\left(t\right)}^{2}-Kt\tau_{1}\left(t\right)-f_{0}\tau_{1}\left(t\right)] \end{array}$$

By substituting Equation (Equation 3) into Equation (Equation 6), the phase of baseband signal after mixing can be further written as
(7)$$\begin{array}{cc} P_{b}\left(t\right) & =2\pi [\frac{1}{2}K{\left(\frac{2(R_{0}-vt)}{c}\right)}^{2}-Kt\frac{2(R_{0}-vt)}{c}-f_{0}\frac{2(R_{0}-vt)}{c}]\\ \; & =-2\pi [\left(K\frac{2R_{0}}{c}-f_{0}\frac{2v}{c}+K\frac{4vR_{0}}{c^{2}}\right)t-\left(K\frac{2v}{c}+K\frac{2v^{2}}{c^{2}}\right)t^{2}+\left(f_{0}\frac{2R_{0}}{c}-K\frac{2R_{0}^{2}}{c^{2}}\right)] \end{array}$$

Since the speed of light c≫v, Equation (Equation 7) can be finally expressed as
(8)$$\begin{array}{cc} P_{b}\left(t\right) & =-2\pi [\left(K\frac{2R_{0}}{c}-f_{0}\frac{2v}{c}\right)t+f_{0}\frac{2R_{0}}{c}]\\ \; & =-2\pi [\left(f_{r}-f_{d}\right)t+f_{0}\frac{2R_{0}}{c}] \end{array}$$
where $f_{r}$ is the frequency offset caused by the target distance and $f_{r}=2KR_{0}/c$ and $f_{d}$ is the frequency offset caused by target speed and $f_{d}=2v/\lambda =2vf_{0}/c$. According to the phase expression of the mixing signal Equation (Equation 8), the mixing signal of a moving target can be written as
(9)$$S_{b}\left(t\right)=A_{1}\cos2\pi \left[\left(f_{r}-f_{d}\right)t+f_{0}\frac{2R_{0}}{c}\right]$$

The distance information in the frequency domain can be acquired and the distance unit can be formed by Fourier transform in each frequency modulation period of the target echo signal received by the radar. On this basis, Doppler frequency information can be further extracted by performing Fourier transform on data points with different periods obtained from the same distance unit.

In order to make the interpretation of the clutter suppression algorithm more accurate, the micro motion characteristic of UAV rotor blades is modeled and analyzed. To simplify the model, the rotor is regarded as an isotropic plate, without considering tilting, bending and twisting. From the perspective of electromagnetic scattering, each blade of a rotor consists of a scattering center, which can be considered as a point with reflectivity. The geometric relationship between the radar and the UAV rotor blades is shown in Figure 1.

In this model, the radar is located at the origin of the Cartesian coordinate system (X,Y,Z). $R_{0}$ is the distance between the radar and the rotor center of the small UAV along the radar LOS when the UAV is in hover state. R(t=0) is the distance between the radar and the top of rotor blade at time 0, and $R(t=t_{0})$ is the distance between the radar and the top of rotor blade at time $t_{0}$. α is the angle between the projection of $R_{0}$ on the XY plane and the *X* axis. β is the angle between $R_{0}$ and the XY plane. Since the UAV is in hover state, the radial velocity of its body relative to the radar can be considered as 0 m/s. To simplify the processing, the parameter is set as $\alpha =\beta =0^{\circ }$. Due to the small detection target and relatively long detection distance, the distance between the rotor tip scattering point and the radar when the UAV hovers at time *t* can be calculated as
(10)$$R\left(t\right)\approx ∥R_{0}+\omega l_{0}{∥}_{2}$$
where ${∥\cdot ∥}_{2}$ is a $l_{2}$-norm or Euclidean norm of a vector, which denotes the square root of the sum of squares of each element in a matrix; $R_{0}$ is the distance between the radar and the rotor center of the hovering UAV; ω is the rotation speed of the rotor blade; $l_{0}$ is the length of UAV’s rotor blade. According to Equation (Equation 9), the difference frequency signal obtained after mixing the transmitted signal with the echo signal received by the LFMCW radar can be expressed as
(11)$$S_{d}\left(t\right)=A_{1}\exp\left\{-j2\pi \left[\frac{K\tau {\left(t\right)}^{2}}{2}-Kt\tau \left(t\right)-f_{0}\tau \left(t\right)\right]\right\}$$
where $A_{1}$ is the amplitude of mixing signal; *K* is the frequency modulation index; τ(t) is the echo delay and τ(t)=2R(t)/c; $f_{0}$ is the carrier frequency. Since τ is represented in fast time dimension, the value of $\tau^{2}$ is small enough to be ignored and the echo signal of a single blade tip on a single rotor received by the radar can be written as
(12)$$S_{p}\left(t\right)=A_{1}\exp\left\{-\frac{j2\pi }{c}\left[-2f_{0}R\left(t\right)-KtR\left(t\right)\right]\right\}$$

By integrating the echo of the entire blade length $l_{0}$, the echo of a single blade on a single rotor can be represented as
(13)$$\begin{array}{cc} S(t) & =\int_{0}^{l_{0}}S_{p}\left(t\right)dr\\ \; & =l_{0}A_{1}sinc\left[2\pi \frac{f_{0}+Kt}{c}l_{0}\cos\left(\omega t\right)\right]\exp\left\{j2\pi \frac{f_{0}+Kt}{c}\left[l_{0}\cos\left(\omega t\right)+2R_{0}\right]\right\} \end{array}$$

If the UAV has *p* rotors, each with *N* blades, and the blades are evenly distributed, then the phase difference of each blade is 2π/N. And the UAV rotor blade echo signal received by the LFMCW radar can be further written as
(14)$$\begin{array}{cc} S_{\Sigma }\left(t\right) & =l_{0}A_{1}sinc\left[2\pi \frac{f_{0}+Kt}{c}l_{0}\cos\left(\omega t+2p\pi /N\right)\right]\\ & \;\exp\left[j2\pi \frac{f_{0}+Kt}{c}l_{0}\cos\left(\omega t+2p\pi /N\right)\right]\exp\left(j2\pi \frac{f_{0}+Kt}{c}2R_{0}\right) \end{array}$$

The conventional LFMCW radar processes the acquired echo signal by using two-dimensional FFT processing to obtain the target distance information and Doppler information, which is an important reference for estimating the UAV rotor speed. In order to directly demonstrate the shortcomings of the traditional high-resolution LFMCW radar in detecting UAVs, the processing of the UAV echo signal received by the radar is simulated and the range-Doppler diagram is obtained. In the simulation, a linear frequency modulation wave with a carrier frequency of 77 GHz is selected as the radar transmission signal to simulate the high resolution E-band millimeter wave radar in reality. The distance between the hovering UAV and the radar is set as 100 m. The length of the UAV rotor is 0.2 m, and the rotational speed of its blade is 83 r/s, which is approaching the actual rotor blade speed of a UAV. The range-Doppler diagram of the UAV echo received by the radar is shown in Figure 2.

According to the setting of simulation parameters, since the length of UAV rotor blade is 0.2 m and the speed is 83 r/s, the maximum radial velocity is 16.6 m/s. Based on the above data, it can be concluded that the theoretical maximum Doppler shift should be around 8521 Hz, but this result cannot be observed from Figure 2, while when the simulation parameter of rotor speed is set very low, such as 10 r/s, this situation will not occur. Therefore, even with the use of a high-resolution radar, the traditional radar signal processing method using range-Doppler to measure rotor blade speed is likely to fail for the small volume UAV target with fast rotation speed. So, it is necessary to find other feasible methods.

### 2.2. Estimation of Rotor Blade Parameters

The rotor blade echo received by the radar appears as a modulated echo in the time domain, which can be described as the flicker phenomenon. The rotation of the rotor changes the distance between the top of the rotor blades and the radar, causing this phenomenon to occur. The time domain representation of rotor blade echo is shown in Figure 3.

The duration of flicker is related to the blade length $l_{0}$ and its rotational rate ω. Within a fixed period, the frequency and interval of the flicker phenomenon are determined by the speed of the UAV rotor, so the Doppler modulation generated by it can be treated as a measurement parameter for UAV identification. Given that the time derivative of the signal phase function is the instantaneous frequency of the signal, the Doppler frequency of the rotor blade echo can be obtained by taking the time derivative of the phase of the received echo signal, which can be represented as
(15)$$\begin{array}{cc} f(t) & =2\pi \left(-\frac{f_{0}+Kt}{c}l_{0}\omega \sin\left(\omega t+2p\pi /N\right)\right)\\ \; & \;+2\pi \left(\frac{K}{c}l_{0}\cos\left(\omega t+2p\pi /N\right)\right) \end{array}$$
where ω is the rotational speed of the UAV rotor; $l_{0}$ is the length of the blade; *N* is the number of blades of each rotor.

From Equation (Equation 15), it can be noted that the micro-Doppler characteristic of the rotor blade echo is modulated by sine and cosine functions. In order to observe the flicker phenomenon clearly, a UAV with a single rotor and double blades is assumed in the simulation demonstration. The rotational rate of double blades of a single rotor is set as ω=4 r/s, and the simulation environment is noise-free. The Doppler modulation is represented in the joint time–frequency domain, which is shown in Figure 4.

As is shown in Figure 4, the first blade and the second blade exhibit four flicker phenomena, respectively, within 1 s, and the rotational speed of the rotor blades can be correctly estimated. It should be noted that, in this paper, we do not consider the influence of the different number of rotor blades, and the number of UAV blades is discussed with the common two blades. Except for the different phases of the waveforms, the echoes of single rotor UAV blades and quad rotor UAV blades have basically the same characteristics in both time and frequency domains. Therefore, the echo characteristics of quad rotor blades can be obtained by an analogy with that of single rotor blades.

## 3. Clutter Suppression Method Based on OMP Algorithm

### 3.1. Sparse Representation

Sparse representation (SR) is a common representation method in human activity recognition, based on the principle that the original signal can be decomposed into a linear combination of finite signals. Therefore, the time-domain echo signal discussed in Section 2 can be discretized for processing in a computer, which can be expressed as
(16)$$y\left(n\right)=\sum_{m=1}^{M}D\left(n,m\right)\alpha \left(m\right)+r\left(n\right)$$
where *n* is the sampling point; *M* is the number of column vectors in the dictionary set, and the column vectors in the dictionary set are also called atoms; D(n,m) is a prior dictionary set required for sparse representation; α(m) is the weight of the corresponding atom in the dictionary set *D*; r(n) is the *n*th sample of the residual. Sparse representation can usually be understood as a simple optimization problem, which can be represented as
(17)$$\min_{\alpha }{∥y-D\alpha ∥}_{2}^{2},\;s.t.{∥\alpha ∥}_{0}\leq \sigma \ll M$$
where ${∥\cdot ∥}_{2}$ is a $l_{2}$-norm or Euclidean norm of a vector, which denotes the square root of the sum of squares of each element in a matrix; ${∥\cdot ∥}_{0}$ is a $l_{0}$-norm of a vector, which expresses the total number of non-zero elements in the vector; σ indicates the sparsity of α; *M* is the number of atoms or columns in D, which should meet the condition M≫σ.

The composition of the dictionary is another key condition for getting SR. As is illustrated in Equation (Equation 17), the quality of the dictionary directly affects the quality of the representation, which determines whether the representation is sparse. This may also result in a specific dictionary being the best for a certain signal and the worst for others. Research has shown that the dictionaries can be obtained based on transformations (kernel functions) [29,30] or through dictionary learning techniques [28].

### 3.2. OMP Dictionary Setting Methods

The construction of a complete dictionary and the proper set of sparsity σ is the major foundation for sparse representation. In order to store the SR of the signals correctly, the condition M>σ must be met. Therefore, a dictionary of at least σ+1 atoms needs to be addressed and bound, which results in a diversity of possible combinations of atoms and the lower error representation of the desired signal. The reality is that it is impossible to establish a dictionary that can maintain compatibility with all types of signals, because the huge amount of information in the dictionary will lead to the high time consumption of the SR lookup, which is not practical. In this section, two OMP dictionary setting methods are proposed to reduce the complexity of the algorithm and the difficulty of setting sparsity, which is suitable for radar detection systems.

The first method originates from the perspective of electromagnetic scattering. After modeling and analyzing the echoes of each scattering point and clutter within the radar detection range, the discrete signals sampled from the echoes of each scattering point and clutter in the time domain is taken as atoms in the dictionary set. Because these echoes are inherently uncorrelated and will still be uncorrelated after being processed by a linear time invariant system, they can be used as a fine dictionary set to sparsely represent the echo signals received by the radar. The disadvantage is that there will be too many scattering points to consider for radar systems with a large detection range, and the generated dictionary with a huge set of information may consume a lot of storage and computing resources. Therefore, it can be concluded that this method is effective when the detection range is small and there are few non-detection targets.

The second dictionary creation method is derived from the multi-channel acquisition idea of an external illuminator radar. Before obtaining the time–frequency spectrum of echoes from UAV rotor blades, a reference channel is set up and used for pre-processing the acquired echoes. From the point of static filtering, the clutter received in the scene, without the target to be tested, can be regarded as the echo of the stationary target, with its distance to the radar antenna and the time delay of each received echo being unchanged. The reference received clutter signal can be obtained by calculating the mean value of clutter signals within several radar scanning cycles, which is also a common method to establish the clutter spectrum. The process of creating a dictionary in this way can be summarized as
(18)$$C\left(n\right)=\frac{1}{N_{s}}\sum_{i=1}^{N_{s}}C\left(n,i\right)$$
where *n* is the number of the sampling point; $N_{s}$ is the total number of the radar scanning cycles; *i* is the number of radar scanning cycles; C(n,i) represents the echo acquired by radar.

It is clear that the second method has less mathematical complexity and consumes fewer resources, and it has been demonstrated to be workable within the engineering error range after numerous trials. If this method is implemented in the dictionary creation process of the OMP algorithm, the complexity of the dictionary information and the difficulty of the sparsity set will be reduced to a certain extent.

According to the specific requirement for the radar detection of UAVs and the characteristic of each approach, the OMP clutter suppression dictionary can be established by comprehensively utilizing the above two methods. As is shown in Figure 5, the theoretical modeling echoes of UAV non-detection points are used to form atoms in the dictionary set D, and the ground clutter collected from the reference channel of the radar in the absence of a UAV is used to form other atoms in the dictionary. When the establishment of dictionary set D is completed, the OMP algorithm can be used to process the echo signals received by the radar in the UAV recognition scenery, and the echo signals of the UAV rotor blades after clutter elimination can be acquired.

### 3.3. Echo Signal Processing Based on OMP Algorithm

Taking the linear equation set y=Dα as an example, if α and D are given in the field of compressed sensing technology, the process of solving y is called compression. While on the other hand, if y and D are given, the process of solving α is called reconstruction. The OMP algorithm implements the process of solving α to a certain extent. It is a greedy algorithm that can be used to recover signals from incomplete coefficients in compressed sensing. The core idea of OMP noise reduction is to recover clutter from the echo signals and eliminate it from the received echoes. The operation description of the OMP algorithm is as follows:(1)First, the dictionary D formed by the clutter and the theoretical modeling echo of UAV non detection points should be provided, as well as the echo signal y received by the radar in a UAV recognition scenario. A STOP_CRITERION is used to stop the algorithm. Typically, the algorithm stops when the residual is less than a certain threshold. In this paper, STOP_CRITERION will stop the algorithm when all the elements in D participate in the calculation.(2)Then, the iteration process should be initialized to begin the search process. Set the initial residual of the original signal r to the original signal. Create the empty sub-matrix $D_{\mathrm{new}}$ and the empty index vector U to register the selected atom and its index in each iteration, respectively. Initialize the counter iteration number j to 0.(3)In the iterative procedure, the maximum absolute inner product $<y\cdot D_{j}>$ should be found, where $D_{j}$ is the *j*th atom of the dictionary D. The inner product is computed to find a part of the signal energy contained in a specific atom. Therefore, the signal can be decomposed atom by atom during the iterative. Then, $D_{j}$ and $\alpha_{j}$ are registered in matrix $D_{\mathrm{new}}$ and U, respectively. The counter iteration *j* increments by one. And the atom $D_{j}$ is removed from dictionary D.(4)The calculation process of updating the residual r is a process of solving the optimization problem, that is, $r={\arg\min}_{\alpha }{∥y-D_{\mathrm{new}}\alpha ∥}_{2}^{2}$. The length of the vector α is equal to the number of non-zero elements of the vector U, which is initialized to the zero vector every time this step is executed. The optimization problem can be solved through the least square method, which can be represented as
(19)$$\alpha =\tilde{D_{\mathrm{new}}^{†}}y$$
where $\tilde{D_{\mathrm{new}}^{†}}$ is the Moore–Penrose pseudo-inverse matrix of $D_{\mathrm{new}}$. Thus, once the pseudo-inverse matrix of $\tilde{D_{j}}$ is found, the sparse contribution vector is obtained with the linear combination between this new matrix and the signal.(5)If the number of iterations is equal to the number of atoms in D, complete the iteration. Otherwise, repeat steps (3) and (4).(6)Iteration is a process of gradually determining the correlation between atoms in a dictionary and the received signal. When the iteration is completed, clutter that does not contain the target detection part of the received signal is removed from the original signal. Finally, the target echo obtained after clutter suppression can be expressed as [31]
(20)$$y^{\prime }\left(n\right)=y\left(n\right)-\Sigma_{j}\alpha_{j}\frac{\tilde{D_{j}}\left(n\right)}{∥\tilde{D_{j}}{\left(n\right)∥}_{2}}$$
where $\alpha_{j}$ is the *j*th element of α.

The pseudocode of the proposed clutter suppression method, based on the OMP algorithm for extracting micro Doppler signals from UAVs, is shown in Algorithm 1.
**Algorithm 1** Clutter suppression method based on OMP algorithmInput: The dictionary formed by the clutter and the theoretical modeling echo of UAV   non–detection points, D; The echo signal received by radar in the UAV identification   scenario, y; The empty matrix used to store sub dictionary, Dnew; The empty vector   used to store the selected atoms and their index number, U;Output: The echo signal after clutter suppression, y′;  1: Calculate the absolute inner product of atom y and Dj;  2: Find the largest absolute inner product αj, register Dj and αj in matrix Dnew and U respectively;  3: **if** length (Dnew) ≤ initial length (D) **then**  4:     Update r and repeat steps 1,2;  5: **else**  6:     Stop iteration;  7: **end if**  8: Filter out Dj from y;  9: **return** y′ and α;

## 4. Simulations and Experiments

### 4.1. Simulations and Analysis

Based on the micro-Doppler echo model established in Section 2, and the clutter suppression method proposed in Section 3, a simulation about the micro-Doppler characteristics extraction of UAV rotor blades is carried out in the simulation environment of MATLAB 2022b (2022. The MathWorks, Inc. USA: Natick, MA.) The weak echo signal from the rotor blades is submerged in the strong ground clutter environment at high friction angle, and the ground clutter distribution generally obeys the Rayleigh distribution model, whose probability density function is
(21)$$f\left(l\right)=\frac{l}{\sigma_{c}^{2}}\exp\left(-\frac{l^{2}}{2\sigma_{c}^{2}}\right),l>0$$
where *l* is the clutter amplitude and $\sigma_{c}$ for its power.

The process of the simulation involves the generation of a target echo signal, the addition of clutter and clutter reduction processing with the OMP algorithm. Finally, the time–frequency spectrums of UAV rotor blades echoes are acquired, from which the characteristics of the UAV rotor blades are extracted. The parameter settings are displayed in Table 1. And the obtained time–frequency spectrums under three different signal-to-noise ratio conditions are shown in Figure 6.

In the case of a low signal-to-noise ratio, the micro-Doppler features of the UAV rotor blades echo’ cannot be clearly displayed, so a Monte Carlo simulation is carried out to examine the performance of the algorithm on various signal-to-noise ratios. As can be seen from Figure 7, with the simulation range of SNR ⊆[−20, 20] dB, the needed SNR to achieve a high detection probability is roughly 10 dB in the absence of a clutter reduction technique. When the OMP-algorithm-based processing method is applied, a high detection probability can still be achieved, even with a low signal-to-noise ratio of −10 dB, which indicates that the proposed clutter suppression algorithm has certain advantages in low signal-to-noise ratio situations.

To observe and compare the Doppler characteristics of echoes with different kinds of UAVs, two kinds of target echo with different parameters are generated in the same simulation scenario. Their rotational speed parameters are set as 30 r/s and 40 r/s, respectively, and their spectrograms in the environment, with a SNR of −5 dB, are shown in Figure 8. It should be noted that the two target positions are set to be the same to simulate the scenario of two different types of UAVs hovering at the same location (the appearance of which is impossible in the actual scene).

The OMP algorithm can reliably separate the blade echo signals of two UAVs from the total received signals, and the rotational speed of the two UAVs’ blades can be determined using the flicker phenomenon of the rotor blades. The received time–frequency spectrum indicates that the flicker phenomenon occurs six times within 0.2 s for a single blade of one of the UAVs, and eight times for a single blade of the other one. According to the analysis above, it can be calculated that the rotational speeds of the two UAVs’ blades are 30 r/s and 40 r/s, respectively, which is consistent with the previous parameter settings.

### 4.2. Field Experiments and Discussion

The field experiments are carried out in an outdoor area with ground clutter, which is a normal flight scene for UAVs. There is no visible electromagnetic or significant wind interference in the environment, and the weather is clear. The transmitted signal of the radar is a linear frequency modulated wave, and the tested UAVs fly and hover without any heavy objects suspended. Here, a linear frequency modulation wave with a bandwidth of 150 MHz is used as the transmission signal of the radar. The pulse repetition time (PRT) is 10 µs, the sampling rate is 25.6 MHz and the sampling time is 1 s. The experiment was conducted on DJI PHANTOM 4PRO, firstly, and the test environment is depicted in Figure 9. It can be noticed that the trees in this scene caused a certain degree of obstruction interference to the detection of the UAV. In this experiment, two types of echo signal data are collected: the data of the environmental clutter without the presence of the UAV and the reflected echo signal received by the radar when conducting UAV detection.

To compare with the processing method proposed in this paper, the EMD algorithm is used to process the echo signal received by the radar, firstly, which decomposes the original signal into a series of intrinsic mode functions (IMF). The time and frequency domain representation of the first four IMFs are shown in Figure 10.

The time domain representation of the main IMFs is shown in the left column, and the frequency domain representation is shown in the right column. As can be seen from the figure, there are serious spectrum broadening and aliasing phenomena, from which it is difficult to obtain the echo signal feature information. The decomposed IMF components are usually retained by EMD through the signal energy threshold. However, the signal energy from clutter and noise is much larger than that from the target in the current experimental scenario, so the target signal cannot be obtained after processing using the EMD algorithm, and the clutter and noise tend to be retained. To supplement the explanation of Figure 10, the Hilbert–Huang transform is performed on the first three IMFs decomposed from the EMD algorithm, and the result is shown in Figure 11. The micro-Doppler characteristics of the UAV rotor blades cannot be obtained from the time–frequency diagram, so the feature parameters required for the recognition cannot be estimated.

Then, we employ the OMP-based clutter suppression approach to process the echo from DJI PHANTOM 4PRO. Because of the periodic characteristics of the processing results, only the shorter time signal processing results are shown in the subsequent spectrogram. The time–frequency spectrums before and after signal processing for the echo of DJI PHANTOM 4PRO is shown in Figure 12.

From Figure 12a, it can be seen that the micro-Doppler signal intensity of the UAV rotor blade echo is weak, which is covered by relatively strong ground clutter. The ground clutter in the experimental scenario is primarily caused by the surrounding buildings, trees and distant soil slopes. Because its frequency is not all zero, the impact of environmental clutter on the received signal cannot be reduced merely by using a high-pass filter. References [32,33] demonstrate that the constant false-alarm rate (CFAR) method can further improve the signal-to-noise ratio after signal processing. Therefore, the OMP algorithm is followed by a two-dimensional CFAR method in the time–frequency domain. Figure 12b shows the time–frequency spectrum of the UAV rotor blade echo obtained after signal processing with the OMP and CFAR algorithms, and the result is clearer and more visible than before. Since different rotors have different elevation angles relative to the radar receiver, the Doppler shift in the frequency domain is still different even if the blades of different rotors have the same velocity.

The focus of this study is to estimate the length and the rotational speed of hovering UAVs’ rotor blades, so the distribution and quantity of rotor blades of UAVs are not currently considered. To obtain the effect shown in Figure 12b, a low-pass filter was used to filter out the spectrum aliasing caused by the other two sets of rotors. Due to the fact that the Doppler frequency shift of the rotor blades relative to the radar receiver is modulated by a sine signal, certain aliasing will still occur in the low frequency band. However, estimating the parameters of the UAV through the flicker phenomenon of the rotor blades in the frequency spectrum can avoid the interference caused by aliasing in calculations.

In order to realize the estimation of rotor blade parameters of different UAVs, DJI PHANTOM 4PRO, DJI Mini 3 and a self-developed UAV based on PIX 2.4.8 open source flight control were selected to participate in the test. The appearance of the three UAVs is shown in Figure 13. And the external field test environment of the experiment on DJI Mini 3 is shown in Figure 14.

Figure 15 and Figure 16 display the time–frequency spectrums of DJI Mini 3 and a self-developed UAV based on PIX 2.4.8 open source flight control, respectively. It can be seen that the flicker phenomenon of rotor blades in the time–frequency spectrum is more clearly visible after the signal processing based on the OMP algorithm. Figure 15 particularly demonstrates the significant clutter suppression effect of this algorithm. The clutter power previously reached a range of −35 dB to −25 dB and this decreases to a range of −50 dB to −40 dB after signal processing. Therefore, it can be concluded that the clutter power can be reduced by 15 dB with the help of the clutter suppression method based on the OMP algorithm.

Through the algorithm proposed in Section 3, the parameter estimation results of three types of UAV can be obtained from Figure 12, Figure 15 and Figure 16, which is shown in Table 2. The theoretical values of the rotor speed parameters are obtained by referring to the data provided by the UAV merchant, and the theoretical value of UAV rotor blade is obtained through actual measurement. When the theoretical values of the two parameters mentioned above are known, the corresponding theoretical values of Doppler frequency shift can be obtained via calculation. In order to minimize the measured error as much as possible, all UAV detection tests are conducted without strong wind interference and with heavy objects suspended. When the radar is used for detection and identification tests, the UAV under test maintains a hovering state. The experimental results indicate that the algorithm used for echo signals processing is helpful for the feature detection and recognition of unknown UAVs. The error between the estimated value and the theoretical value is usually within 5%. Due to the algorithm’s excellent performance in clutter suppression, the micro-Doppler features of UAVs are easily visible in the time–frequency spectrum, allowing for the estimation of parameters such as rotor blade speed and blade length.

After extracting the characteristic parameters of different kinds of UAVs, the mean clustering algorithm can be used to classify and identify the unknown UAV. For example, the commonly used K-means algorithm is suitable for this scenario. The loss function of the unknown UAV numbered *m* relative to the typical UAV numbered *n* can be defined as $J_{m,n}=\sum_{i=1}^{F}{∥I_{m}\left(i\right)-T_{n}\left(i\right)∥}^{2}$, where *F* is the total number of feature parameters; $I_{m}\left(i\right)$ is the *i*-th characteristic parameter of the unknown UAV numbered *m*; $T_{n}\left(i\right)$ is the *i*-th characteristic parameter of a typical UAV numbered *n*. If there is $\arg\min_{n}J_{m,n}$, the detected UAV with corresponding numbered *m* can be considered as a typical UAV with known model numbered *n*.

## 5. Conclusions

In this study, a clutter suppression method based on an OMP algorithm is proposed to extract the micro-Doppler signature from the echo signals of the hovering UAV received by a LFMCW radar. The method, based on sparse representation, establishes a complete dictionary, firstly, and effectively achieves clutter suppression in complex environments with low resource consumption and computational complexity. In addition, by filtering the strong clutter in the echo signals, this approach can make up for the defect of conventional algorithms such as EMD, which cannot detect the target due to the small energy of the target echo. Based on the obtained time–frequency spectrums of echo signals after processing, the parameters of blade length and rotor rotational velocity can be calculated. It should be noted that, when the blade speed of the tested UAV is too high, the parameter estimation effect of this method will decrease. The reason is that the sinusoidal waveform interval presented by the multi-rotor speed is too small in the time–frequency domain, and the identification cannot be accurately completed due to the failure to distinguish the respective waveform peaks of different blades. The radar detection experiments of three different types of UAVs were carried out and the estimated parameters of UAVs blades were basically consistent with those provided by manufacturers, which demonstrates the potential application of the proposed method in assisting UAV identification and classification in complex environments such as those with buildings and trees. The research provides a certain prior support for subsequent targeted UAV monitoring or UAV counter-experiments.

## Figures and Tables

**Figure 1 sensors-23-07922-f001:**
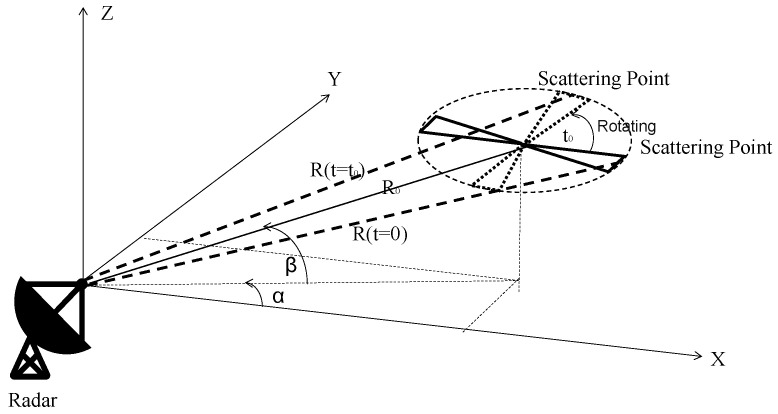
The geometric relationship between the radar and the UAV rotor blades.

**Figure 2 sensors-23-07922-f002:**
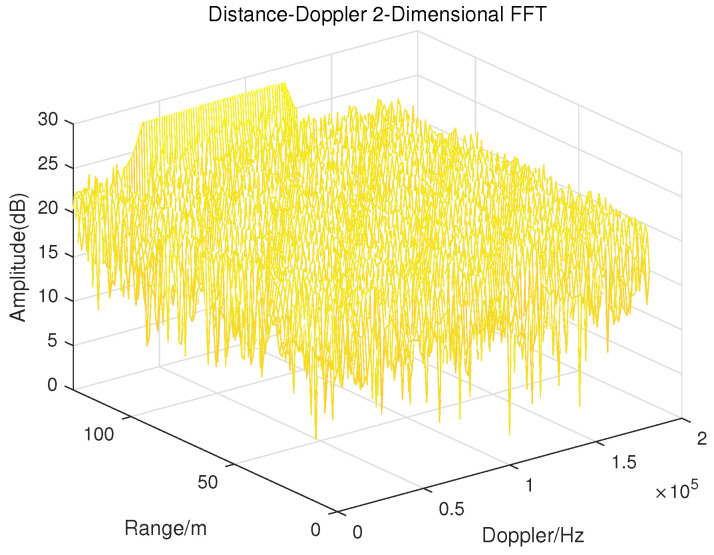
R-D diagram of echoes received by LFMCW radar in the simulation.

**Figure 3 sensors-23-07922-f003:**
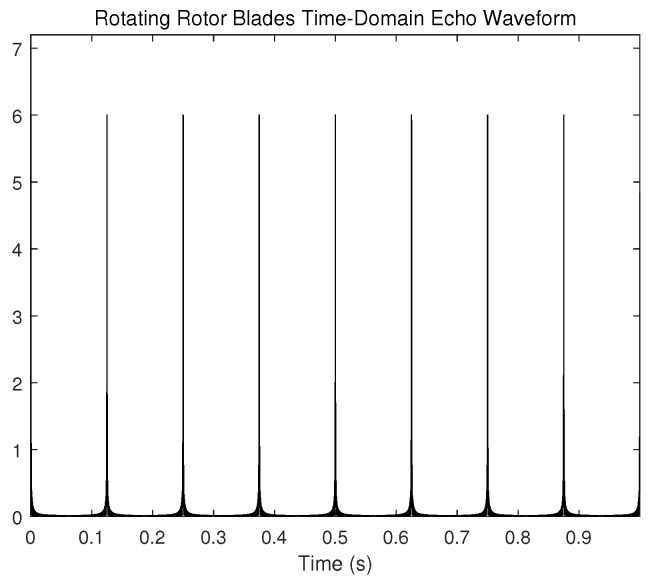
Flicker phenomenon of the rotor blade echo in time domain.

**Figure 4 sensors-23-07922-f004:**
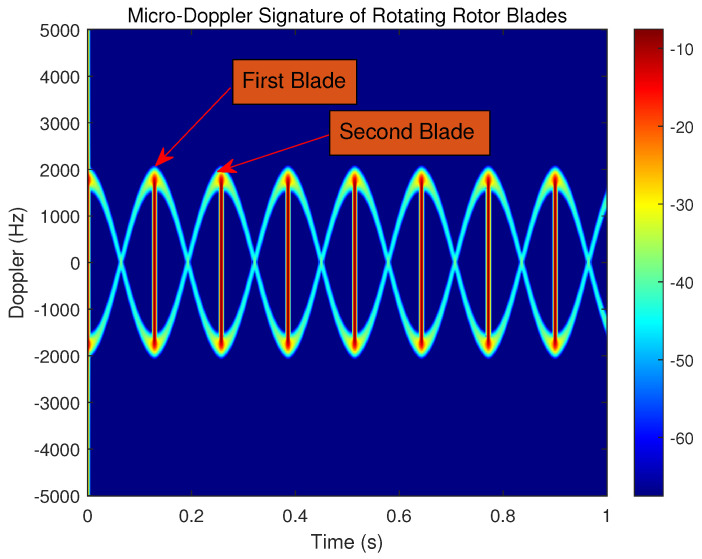
Time–frequency spectrum of rotor blade echo.

**Figure 5 sensors-23-07922-f005:**
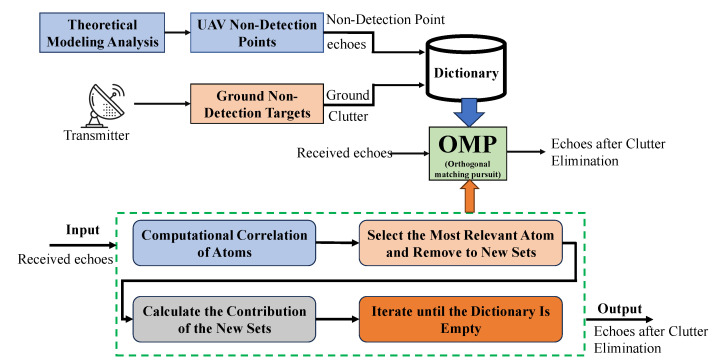
A clutter suppression method based on OMP algorithm in radar systems.

**Figure 6 sensors-23-07922-f006:**
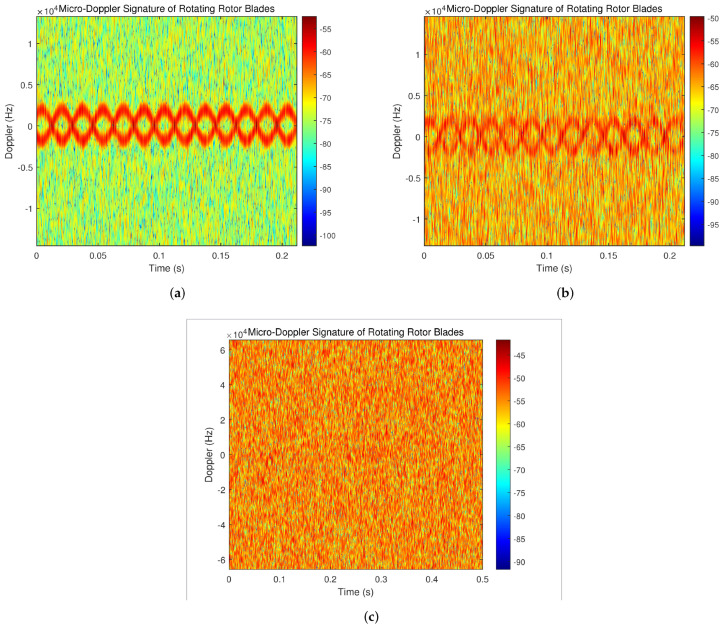
Time–frequency spectrum of the UAV rotor blade echoes under different signal-to-noise ratios. (**a**) SNR = −5 dB. (**b**) SNR = −10 dB. (**c**) SNR = −15 dB.

**Figure 7 sensors-23-07922-f007:**
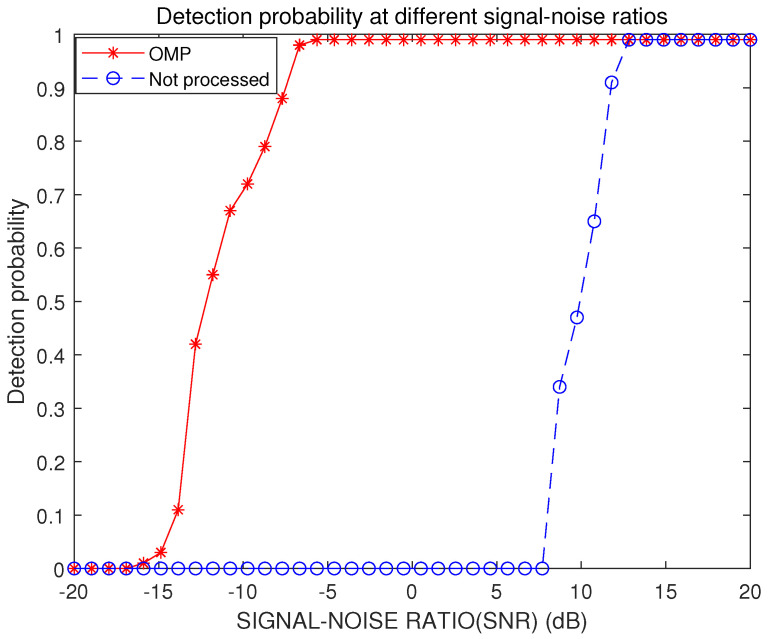
Detection probability under different signal–to–noise ratios before and after clutter suppression processing.

**Figure 8 sensors-23-07922-f008:**
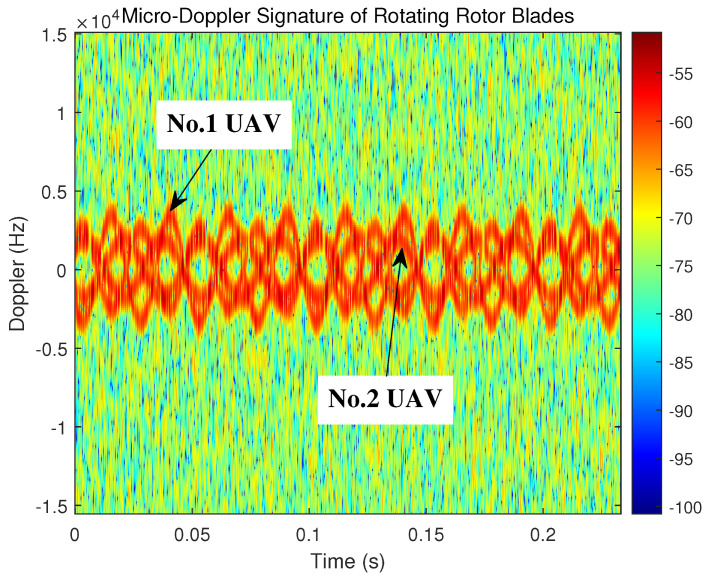
Time–frequency spectrum of the echoes from different types of rotor blade.

**Figure 9 sensors-23-07922-f009:**
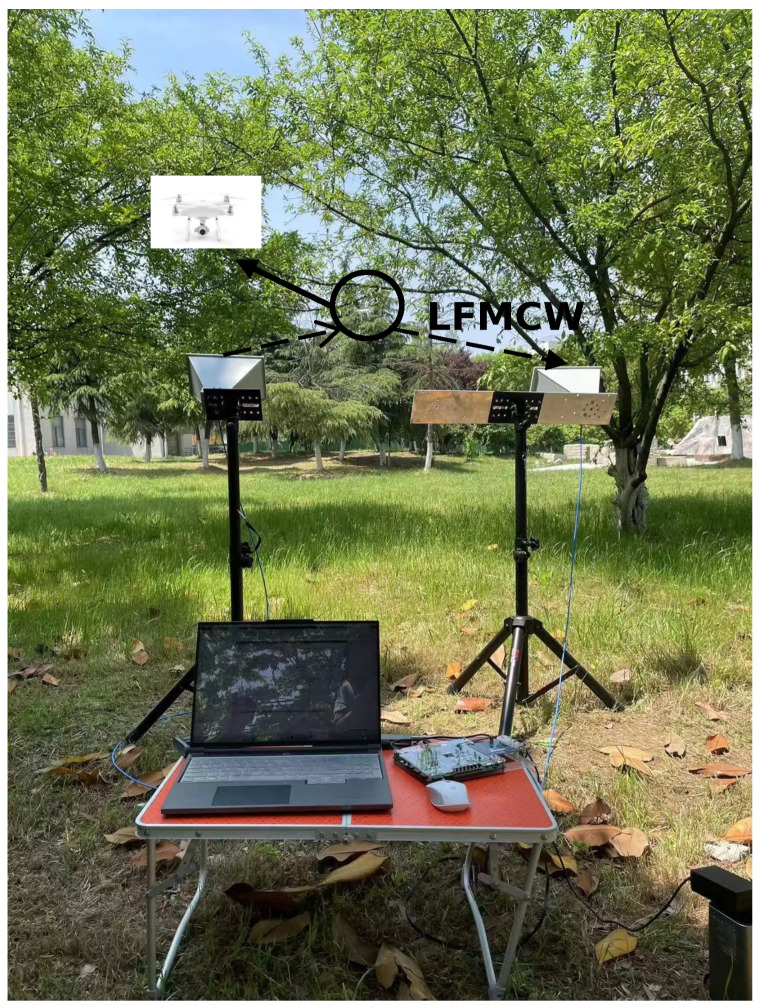
The test scene for DJI PHANTOM 4Pro detection. The transmission signal is LFMCW signal.

**Figure 10 sensors-23-07922-f010:**
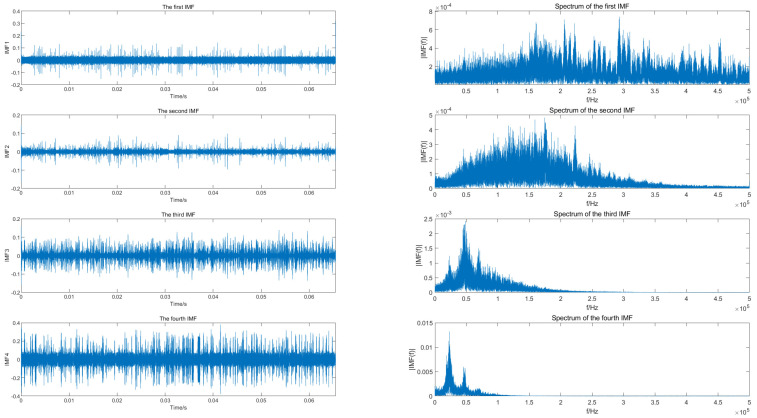
Time–domain and frequency–domain reprensentations of the main IMFs components after EMD decomposition of DJI PHANTOM 4Pro measured data.

**Figure 11 sensors-23-07922-f011:**
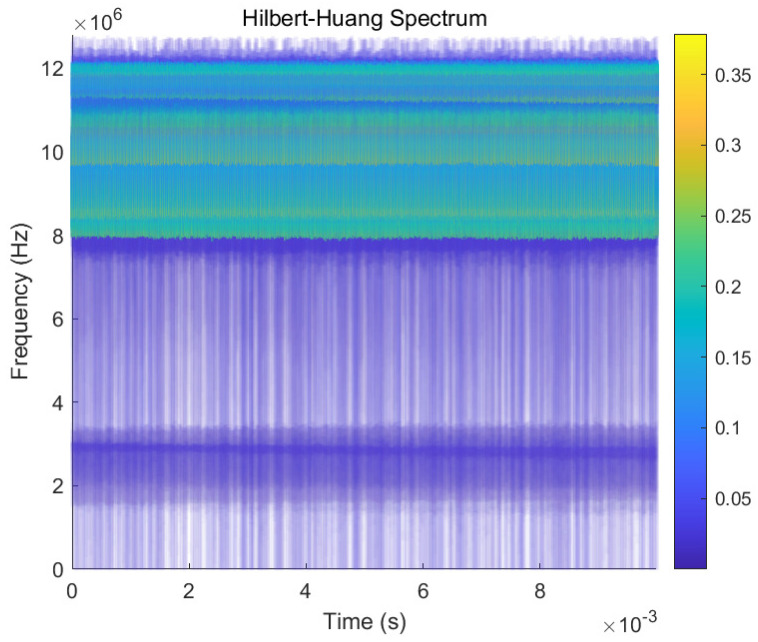
Hilbert–Huang transform for the first three IMFs decomposed by EMD.

**Figure 12 sensors-23-07922-f012:**
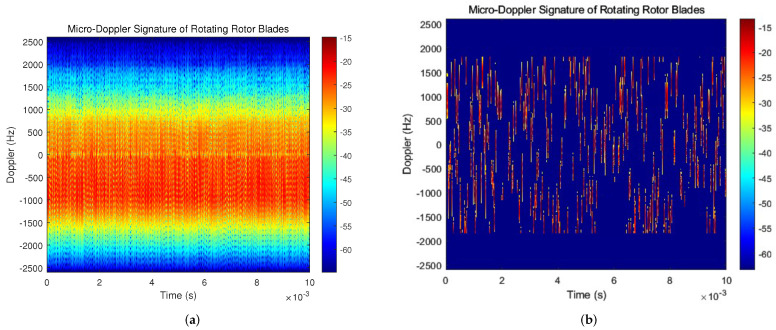
Time–frequency spectrum before and after signal processing for DJI PHANTOM 4PRO. (**a**) Time–frequency spectrum of received echo signal. (**b**) Time–frequency spectrum processed using OMP and CFAR algorithm.

**Figure 13 sensors-23-07922-f013:**
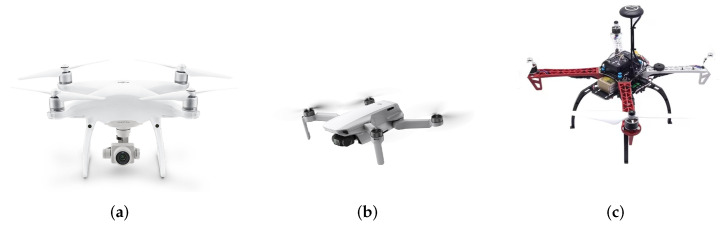
The appearance of the three UAVs involved in the trial. (**a**) The appearance of DJI PHANTOM 4PRO. (**b**) The appearance of DJI Mini 3. (**c**) The appearance of UAV based on PIX2.4.8 open source flight control.

**Figure 14 sensors-23-07922-f014:**
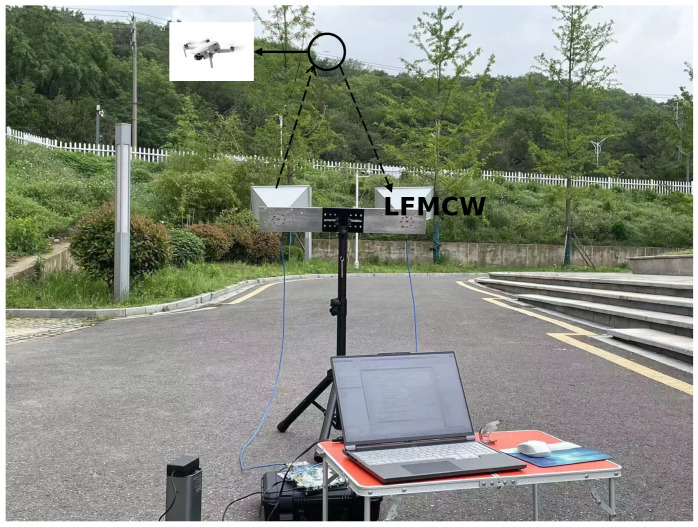
The test scene for DJI Mini 3 detection.

**Figure 15 sensors-23-07922-f015:**
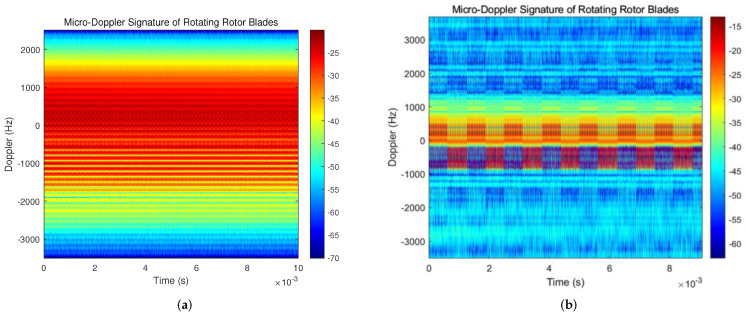
Time–frequency spectrum before and after signal processing for DJI Mini 3. (**a**) Time–frequency spectrum of received echo signal. (**b**) Time–frequency spectrum processed by OMP algorithm.

**Figure 16 sensors-23-07922-f016:**
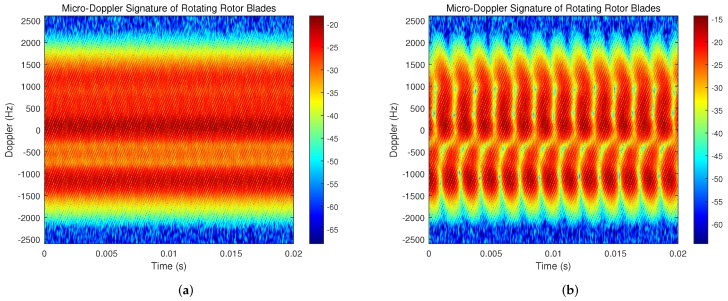
Time–frequency spectrum before and after signal processing of a self–developed UAV based on PIX2.4.8 open–source flight control. (**a**) Time–frequency spectrum of received echo signal. (**b**) Time–frequency spectrum processed by OMP algorithm.

**Table 1 sensors-23-07922-t001:** Simulation parameter settings.

Parameter	Value or Description
Cartesian coordinate of radar (unit 1 m)	[0, 0, 0]
Cartesian coordinate of UAV (unit 1 m)	[100, 100, 0]
Radar carrier frequency $f_{c}$	5 GHz
Number of UAV blades *N*	2
UAV blade rotational speed ω	30 r/s
UAV blade length $l_{0}$	0.3 m
UAV motion status	hover
SNR of received signal	(−5 dB, −10 dB, −15 dB)

**Table 2 sensors-23-07922-t002:** Result of the micro-motion parameter estimation for the quadcopter hovering UAVs.

	DJI PHANTOM 4PRO		
**Feature Parameters**	**Theoretical**	**Estimated**	**Deviation**
Doppler shift	1913.7 Hz	1843.4 Hz	3.67%
Rotor rotational speed	107.0 r/s	101.3 r/s	5.33%
Blade length	120.0 mm	122.1 mm	1.75%
	DJI Mini 3		
Feature parameters	Theoretical	Estimated	Deviation
Doppler	1346.0 Hz	1370.3 Hz	1.8%
Rotor rotational speed	100.7 r/s	100 r/s	0.7%
Blade length	70.0 mm	67.3 mm	3.9%
	a UAV based on PIX2.4.8		
Feature parameters	Theoretical	Estimated	Deviation
Doppler shift	2210.6 Hz	2180.2 Hz	1.37%
Rotor rotational speed	80.0 r/s	83.3 r/s	4.12%
Blade length	132.0 mm	125.0 mm	6.30%

## Data Availability

The data presented in this study are available on request from the corresponding author.

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
