# Peer review of "Micro-Doppler Signature Detection and Recognition of UAVs Based on OMP Algorithm"

_sensors, 2023, doi:10.3390/s23187922_

Round 1
Reviewer 1 Report
1. Define acronym before citing them
2. Improve on Figure 10
3. Improve on the expression of statement
Moderate editing of English language is required
Author Response
We thank you very much for giving us an opportunity to revise our manuscript, and we also appreciate reviewers very much for your positive and constructive comments and suggestions on our manuscript.
Comments 1: Define acronym before citing them.
Response 1: We've gone over the full text to make sure that each abbreviated word has a full explanation when it first appears.
Comments 2: Improve on Figure 10.
Response 2: We supplement the Hilbert-Huang Transform (HHT) algorithm widely used in the academic circle, convert the time domain signal to the time domain as well, observe the "flicker" phenomenon mainly relied on for its estimation, and then compare with the algorithm proposed in this paper, it can be seen that the information about the target in the time-frequency graph is not available. We added control treatment to highlight the innovation and effect of the algorithm in this paper, so as to state our conclusions more clearly.
Comments 3: Improve on the expression of statement.
Response 3: The English language in the revised manuscript has been carefully corrected to improve grammar and readability. Errors in detail about units and the problem of missing spaces have been corrected. In addition, we have rewritten the abstract and conclusion, in order to make the sentences more smooth and more able to show the innovation points and application scenarios of our article.
Reviewer 2 Report
The introduction is accurate.
In the first row and second row, which is the unit measurement for table 1.
The figure 10 is not clear. I suggest expanding the graphics and superpositioning the figures to compare them.
In table 2 I think that you analize in deep the results. The 'blade length' row, for instance, has a percent deviation that is not linear in relation to the measurement of the blade length. It’s strange that it’s not growing or descending linearly if the blade length grows.
The conclusions must be rewritten
Author Response
We thank you very much for giving us an opportunity to revise our manuscript, and we also appreciate reviewers very much for your positive and constructive comments and suggestions on our manuscript.
Comments 1: In the first row and second row, which is the unit measurement for table 1.
Response 1: The assumption of the whole scenario is established in a spatial cartesian coordinate system in units of ‘m’. At the same time, we have reviewed the full-text table and added missing unit measurement in rows and columns and gave an explanation.
Comments 2: The figure 10 is not clear. I suggest expanding the graphics and superpositioning the figures to compare them.
Response 2: We supplement the Hilbert-Huang Transform (HHT) algorithm widely used in the academic circle, convert the time domain signal of first three IMFs(Intrinsic Mode Function, IMF) to the time domain as well, observe the "flicker" phenomenon mainly relied on for its estimation, and then compare with the algorithm proposed in this paper, it can be seen that the information about the target in the time-frequency graph is not available. We added control treatment to highlight the innovation and effect of the algorithm in this paper, so as to state our conclusions more clearly.
Comments 3: In table 2 I think that you analize in deep the results. The 'blade length' row, for instance, has a percent deviation that is not linear in relation to the measurement of the blade length. It’s strange that it’s not growing or descending linearly if the blade length grows.
Response 3: First of all, we agree with you on this point. The relationship between estimation error and estimation parameters is not the main purpose of this paper. According to the previous research results of scholars such as V. C. Chen, in fact, large slow targets are often less difficult to detect, and the accuracy is theoretically more accurate. However, in this paper, different small UAVs are mainly placed in different complex environments for micro-Doppler detection and estimation recognition. Therefore, the impact of environment, UAV mechanical structure and other factors on the estimation error significantly exceeds the impact of blade length between different UAVs. The purpose of this paper is to propose a method that can be applied to clutter suppression and micro-Doppler estimation in most complex environments. Some relationships between estimation errors and estimation parameters themselves are not deeply discussed. Thank you for your constructive questions, which have proposed a new idea for our future research.
Comments 4: The conclusions must be rewritten.
Response 4: We rewrote the conclusion of the paper, and compared with the original version, we emphasized the innovation and application prospect of the algorithm. In this paper, the theoretical basis of the proposed method is introduced in detail, the general steps of the processing of the paper are introduced, and the results of comparison with the widely used EMD algorithm are written. At the same time, some limitations of the algorithm are also written, that is, the target rotor speed cannot exceed a threshold, which is preliminarily estimated to be between 110r/s and 120r/s through experiments. The specific conclusions are rewritten as follows: In this study, a clutter suppression method based on OMP algorithm is proposed to extract the micro-Doppler signature from the echo signals of the hovering UAV received by a LFMCW radar. The method based on sparse representation establishes a complete dictionary firstly, and effectively achieves the clutter suppression in complex environment with low resource consumption and computational complexity. In addition, by filtering the strong clutter in the echo signals, this approach can make up for the defect of conventional algorithms such as EMD, which cannot detect the target due to the small energy of the target echo. Based on the obtained time-frequency spectrums of echo siganls after processing, the parameters of blade length and rotor rotational velocity can be calculated. It should be noted that when the blade speed of the tested UAV is too high, the parameter estimation effect of this method will decrease. The reason is that the sinusoidal waveform interval presented by the multi-rotor speed is too small in the time-frequency domain, and the identification cannot be accurately completed due to the failure to distinguish the respective waveform peaks of different blades. The radar detection experiments of three different types of UAVs were carried out and the estimated parameters of UAVs blades were basically consistent with those provided by manufacturers, which demonstrates the potential application of the proposed method in assisting UAV identification and classification in complex environments such as buildings and trees. The research provides a certain prior support for subsequent targeted UAV monitoring or UAV counter-experiments.
Reviewer 3 Report
1. What is the main question addressed by the research?
The paper is dedicated to micro-Doppler signature detection and recognition of unmanned aerial vehicles. The method applied is based on orthogonal matching pursuit algorithm.
2. Do you consider the topic original or relevant in the field, and if so, why?
The topic is original and relevant in the field. The results obtained show that the method considered can effectively reduce clutter interference and aid in the extraction of micro-Doppler information from unknown unmanned aerial vehicles.
3. What does it add to the subject area compared with other published material?
The authors have proposed a clutter suppression method based on orthogonal matching pursuit algorithm to process the echo signals of the hovering unmanned aerial vehicles received by radar. Compared with traditional methods, it can achieve relatively accurate estimation of unmanned aerial vehicle micro-motion parameters under low signal-to-noise ratio condition. Field experiments show that this method can reduce the clutter power effectively. The rotor speed and blade length of the unmanned aerial vehicles can be calculated from the time-frequency spectrums of the unmanned aerial vehicle blade echo, which provides a part of the basis for the subsequent identification of unknown unmanned aerial vehicles.
4. What specific improvements could the authors consider regarding the methodology?
There is no need to make any improvements or something else. The authors outlined shortly the current achievements in the field in the introduction, and provided detailed description of the materials and method, as well as the obtained results, their discussion, and conclusions.
5. Are the conclusions consistent with the evidence and arguments presented and do they address the main question posed?
The conclusions are consistent with the evidence and arguments presented in the manuscript and address the main questions of their study. But section Conclusions should be extended. Please, also provide the information about possible applicability of the results obtained.
6. Are the references appropriate?
The references are appropriate.
7. Please include any additional comments on the tables and figures.
All the tables and figures are appropriate. They show well the research and experiment details and results. Please, increase resolution in Figure 2, and enlarge Figure 6, 10, 14, and 15. Please, use black color instead of read for arrows and text in Figure 9, and 13.
8. Other comments.
The paper should have the following structure (see the journal template): Introduction, Materials and Methods, Results, Discussion, and Conclusions. Please, reorganize your paper in accordance with this structure.
Please, add in Abstract the information on where the results obtained can be used.
Lines 46 and 49. Please, add a blank between the word and following reference in the text.
Line 188. As ‘No.1 and No.2 blade’ are the same as ‘the first blade and the second blade’, use them in accordance with the Figure 4.
Please, abbreviate journal names, and provide missing DOIs in references.
After detailed consideration of the manuscript, I have found that the results obtained are new and significant for the field. The manuscript is written mostly well, but still needs some corrections before its publication.
So, the paper needs a minor revision.
Author Response
We thank you very much for giving us an opportunity to revise our manuscript, and we also appreciate reviewers very much for your positive and constructive comments and suggestions on our manuscript.
Comments 1: Section Conclusions should be extended.
Response 1: We rewrote the conclusion of the paper, and compared with the original version, we emphasized the innovation and application prospect of the algorithm. In this paper, the theoretical basis of the proposed method is introduced in detail, the general steps of the processing of the paper are introduced, and the results of comparison with the widely used EMD algorithm are written. At the same time, some limitations of the algorithm are also written, that is, the target rotor speed cannot exceed a threshold, which is preliminarily estimated to be between 110r/s and 120r/s through experiments. The specific conclusions are rewritten as follows: In this study, a clutter suppression method based on OMP algorithm is proposed to extract the micro-Doppler signature from the echo signals of the hovering UAV received by a LFMCW radar. The method based on sparse representation establishes a complete dictionary firstly, and effectively achieves the clutter suppression in complex environment with low resource consumption and computational complexity. In addition, by filtering the strong clutter in the echo signals, this approach can make up for the defect of conventional algorithms such as EMD, which cannot detect the target due to the small energy of the target echo. Based on the obtained time-frequency spectrums of echo siganls after processing, the parameters of blade length and rotor rotational velocity can be calculated. It should be noted that when the blade speed of the tested UAV is too high, the parameter estimation effect of this method will decrease. The reason is that the sinusoidal waveform interval presented by the multi-rotor speed is too small in the time-frequency domain, and the identification cannot be accurately completed due to the failure to distinguish the respective waveform peaks of different blades. The radar detection experiments of three different types of UAVs were carried out and the estimated parameters of UAVs blades were basically consistent with those provided by manufacturers, which demonstrates the potential application of the proposed method in assisting UAV identification and classification in complex environments such as buildings and trees. The research provides a certain prior support for subsequent targeted UAV monitoring or UAV counter-experiments.
Comments 2: Please, also provide the information about possible applicability of the results obtained.
Response 2: We have added some scenarios and possible uses for the results to the summary. “The method proposed in this paper can complete the identification and classification of unknown UAVs with high accuracy in the presence of a large number of complex environments such as buildings and trees at low altitude. The identification and classification of low-altitude small hovering UAVs also provides certain a priori conditions for the subsequent targeted UAV control in the civilian field or the interference and confrontation of UAV systems in the military field.”
Comments 3: Increase resolution in Figure 2, and enlarge Figure 6, 10, 14, and 15. Please, use black color instead of read for arrows and text in Figure 9, and 13.
Response 3: We have increased resolution in Figure 2, and enlarge Figure 6, 10, 14, and 15 and used black color instead of read for arrows and text in Figure 9, and 13.
Comments 4: The paper should have the following structure (see the journal template): Introduction, Materials and Methods, Results, Discussion, and Conclusions. Please, reorganize your paper in accordance with this structure.
Please, add in Abstract the information on where the results obtained can be used.
Lines 46 and 49. Please, add a blank between the word and following reference in the text.
Line 188. As ‘No.1 and No.2 blade’ are the same as ‘the first blade and the second blade’, use them in accordance with the Figure 4.
Please, abbreviate journal names, and provide missing DOIs in references.
Response 4: We have corrected all the issues mentioned above, and the changes are highlighted in red in the text. Thanks for pointing out the irregularities in our article.
Reviewer 4 Report
Abstract
Abbreviations must be defined in the abstract and in the rest of the paper. How we define an abbreviation, e.g., Unmanned Aerial Vehicle (UAV), please check that all used abbreviations in the paper are defined correctly.
The abstract must summarise the suggested approach.
Introduction
It is beneficial to talk about the limitations of using UAVs in urban areas, The following references may help you:
McTegg, S.J., Tarsha Kurdi, F., Simmons, S., Gharineiat, Z. 2022. Comparative approach of unmanned aerial vehicle restrictions in controlled airspaces. Remote Sens. 2022, 14, 822. https://doi.org/10.3390/rs14040822.
Stöcker, C.; Bennett, R.; Nex, F.; Gerke, M.; Zevenbergen, J. Review of the Current State of UAV Regulations. Remote Sens. 2017, 9, 459. https://doi.org/10.3390/rs9050459.
Line 30: Please remove “and so on”.
Please highlight the novelty and the contribution of the paper.
Micro-Doppler Echo Analysis
Please don’t put two section titles consecutively, you must add a transition paragraph between them, Please check all the paper.
Micro-Doppler Echo Model
Please cite the used equations in the text, e.g., in Line 89, transmission cycle can be represented in the time domain (Equation 1). Please check all equations.
If you develop any equation, please explain how you develop it, if not, please add reference(s) (please check all equations in the paper).
Please define all Equation parameters next to the equation, you can provide a numeric example about the calculation if that makes it clearer.
Please add an extended flowchart that explains the suggested approach steps.
Clutter Suppression Based on OMP Algorithm
Parameters and abbreviations must be defined in Figure 5. Figure 5 must be extended to be more comprehensive.
Input and output must be highlighted in Algorithm 1.
Please avoid using (we, our, and us), use the passive voice. Please check the paper.
Conclusion
Please analyze the limitations of the suggested approach.
Minor editing of English language required.
Author Response
We thank you very much for giving us an opportunity to revise our manuscript, and we also appreciate reviewers very much for your positive and constructive comments and suggestions on our manuscript.
Comments 1: Abbreviations must be defined in the abstract and in the rest of the paper. How we define an abbreviation, e.g., Unmanned Aerial Vehicle (UAV), please check that all used abbreviations in the paper are defined correctly.
Response 1: We've gone over the full text to make sure that each abbreviated word has a full explanation when it first appears.
Comments 2: The abstract must summarise the suggested approach.
Response 2: We have summarised the suggested approach in the abstract: In this paper, for hovering UAV, an innovative clutter suppression method based on sparse representation, Orthogonal Matching Tracking (OMP) algorithm, is proposed to create a complete dictionary set of ambient clutter, and then efficiently suppress clutter from the received signal. Then the suppressed signals are analyzed in the time-frequency domain. By analyzing the flicker phenomenon of UAV rotor blades and micro-Doppler characteristics, the characteristic parameters of unknown UAV models are estimated and identified.
Comments 3: It is beneficial to talk about the limitations of using UAVs in urban areas.
Response 3: In the introduction of the article, we have added some explanations of the necessity of this technology, that is, some restrictions on the use of drones in cities, and the serious consequences caused by illegal drones, and added some literature to testify our views and application prospects.
Comments 4: Some questions about article layout and formatting. Line 30: Please remove “and so on”. Please don’t put two section titles consecutively, you must add a transition paragraph between them, Please check all the paper. Input and output must be highlighted in Algorithm 1. Please avoid using (we, our, and us), use the passive voice. Please check the paper.
Response 4: We have corrected all the issues mentioned above, and the changes are highlighted in red in the text. Thanks for pointing out the irregularities in our article. In this paper, we have realized the emphasis on input and output by bolding and enlarging the input and output variables.
Comments 5: Please highlight the novelty and the contribution of the paper.
Response 5: We have rewritten the abstract and conclusion, which briefly describes the basic process of our proposed method, highlight the novelty and the contribution of the paper. The content emphasizes in it that compared with the existing methods, our innovation lies in proposing a new method to establish a complete dictionary, reducing the consumption of resources, and proposing a clutter suppression method and time-frequency analysis method based on matched orthogonal tracking theory. It can accomplish the purpose that conventional methods cannot accomplish in complex environment. And our method has a wide range of application scenarios, and provides certain conditions for the follow-up research.
Comments 6: Please cite the used equations in the text, e.g., in Line 89, transmission cycle can be represented in the time domain (Equation 1). Please check all equations. If you develop any equation, please explain how you develop it, if not, please add reference(s) (please check all equations in the paper). Please define all Equation parameters next to the equation, you can provide a numeric example about the calculation if that makes it clearer.
Response 6: We have checked the formulas in the paper, and quoted the formulas obtained from other literatures. The formulas derived from the paper give detailed procedures and explanations. And the parts have been revised are pointing out the irregularities in our article. By checking, we introduce the physical meaning and defining symbols of variables in all formulas, where they first appear. As for the part of numerical examples, the paper gives the Settings of some parameters in the following specific simulation and test, and these parameters can be replicated under normal circumstances.
Comments 7: Please add an extended flowchart that explains the suggested approach steps. Parameters and abbreviations must be defined in Figure 5. Figure 5 must be extended to be more comprehensive.
Response 7: We have redrawn Figure 5 to write the full name of the abbreviations and use arrows to illustrate the processing flow. At the same time, the specific OMP algorithm processing steps are extended, which are mainly divided into four steps and illustrated in the figure. The specific four steps of how to process signals are also detailed in the following article.
Comments 8: Conclusion. Please analyze the limitations of the suggested approach.
Response 8: We rewrote the conclusion of the paper, and compared with the original version, we emphasized the innovation and application prospect of the algorithm. In this paper, the theoretical basis of the proposed method is introduced in detail, the general steps of the processing of the paper are introduced, and the results of comparison with the widely used EMD algorithm are written. At the same time, some limitations of the algorithm are also written, that is, the target rotor speed cannot exceed a threshold, which is preliminarily estimated to be between 110r/s and 120r/s through experiments. The specific conclusions are rewritten as follows: In this study, a clutter suppression method based on OMP algorithm is proposed to extract the micro-Doppler signature from the echo signals of the hovering UAV received by a LFMCW radar. The method based on sparse representation establishes a complete dictionary firstly, and effectively achieves the clutter suppression in complex environment with low resource consumption and computational complexity. In addition, by filtering the strong clutter in the echo signals, this approach can make up for the defect of conventional algorithms such as EMD, which cannot detect the target due to the small energy of the target echo. Based on the obtained time-frequency spectrums of echo siganls after processing, the parameters of blade length and rotor rotational velocity can be calculated. It should be noted that when the blade speed of the tested UAV is too high, the parameter estimation effect of this method will decrease. The reason is that the sinusoidal waveform interval presented by the multi-rotor speed is too small in the time-frequency domain, and the identification cannot be accurately completed due to the failure to distinguish the respective waveform peaks of different blades. The radar detection experiments of three different types of UAVs were carried out and the estimated parameters of UAVs blades were basically consistent with those provided by manufacturers, which demonstrates the potential application of the proposed method in assisting UAV identification and classification in complex environments such as buildings and trees. The research provides a certain prior support for subsequent targeted UAV monitoring or UAV counter-experiments.
Comments 9: Minor editing of English language required.
Response 9: The English language in the revised manuscript has been carefully corrected to improve grammar and readability. Errors in detail about units and the problem of missing spaces have been corrected.